# Mobile health supported real-time guidance and debriefing for newborn resuscitation: A pilot study of LIVEBORN feedback

Kourtney R. Bettinger[1], Daniel K. Ishoso[2], Amy S. Mackay[3], Carl L. Bose[3], Patricia P. Gomez[4], Ingunn A. Haug[5], Helge Myklebust[5], Benjamin H. Chi[6], Eric M. Mafuta[2], Jackie K. Patterson[3]*

1 Department of Pediatrics, University of Kansas, Kansas City, Kansas, United States of America,
2 School of Public Health, University of Kinshasa, Kinshasa, Democratic Republic of Congo,
3 Department of Pediatrics, University of North Carolina at Chapel Hill, Chapel Hill, North Carolina, United States of America, 4 Jhpiego, Baltimore, Maryland, United States of America, 5 Laerdal Medical, Stavanger, Norway, 6 Department of Obstetrics and Gynecology, University of North Carolina at Chapel Hill, Chapel Hill, North Carolina, United States of America

* jackie_patterson@med.unc.edu

## Abstract

Basic resuscitation practices, particularly bag-mask ventilation, reduce newborn deaths from respiratory depression. There is strong scientific premise for improving bag-mask ventilation with feedback strategies, but there are significant barriers to bedside feedback in low-resource settings. To address these barriers, we developed LIVEBORN, a mobile health application to support feedback for newborn resuscitations. LIVEBORN uses data on provider actions and the newborn's condition entered in real-time by an observer to provide real-time guidance *during* resuscitation and support debriefing *after* resuscitation. In a pilot study, we designed and then evaluated strategies to incorporate LIVEBORN Feedback into clinical practice at two health facilities in the Democratic Republic of the Congo, with one facility allocated to real-time guidance and one to debriefing. Providers at each facility used a participatory research methodology called Trials of Improved Practices to design and refine their strategy prior to pilot testing. The primary outcome of the pilot study was the feasibility of observing resuscitation care with LIVEBORN, defined as the percentage of births observed using LIVEBORN with a threshold of at least 50% of births observed to achieve feasibility. We also evaluated usability with the System Usability Scale and explored midwives' perceptions of LIVEBORN Feedback in focus group discussions. During the pilot test, we found both strategies to be feasible with 74% of births observed with LIVEBORN at the real-time guidance facility and 67% at the debriefing facility. The strategy was also sufficiently usable with a System Usability Scale median score of 68 (Q1 65, Q3 78). Midwives perceived LIVEBORN Feedback to be helpful and believed it could save lives, but sometimes disagreed with LIVEBORN Feedback's guidance to ventilate. In conclusion, we identified context-specific,

**Data availability statement:** All data files are available from the UNC Dataverse database (https://dataverse.unc.edu/dataset.xhtml?persistentId=doi:10.15139/S3/HQOLY3).

**Funding:** KRB: UG1 OD024943 - National Institute of General Medical Sciences; https://nam12.safelinks.protection.outlook.com/?url=https%3A%2F%2Fwww.nigms.nih.gov%2FResearch%2FDRC-B%2FIDeA%2FPages%2Fdefault.aspx-&data=05%7C02%7C%7Ced77110b-c0e443962aea08de4438e926%7C58b3d54f16c942d3af081fcabd095666%7C1%7C0%7C639023208546328989%-7CUnknown%7CTWFpbGZsb3d8eyJFbX-B0eU1hcGkiOnRydWUsIlYiOiIwLjAuMDAw-MCIsIlAiOiJXaW4zMiIsIkFOIjoiTWFpbCIs-IldUIjoyfQ%3D%3D%7C0%7C%7C%7C%7C&s-data=QVR1qa6kaX0VuKY%2Be2AUvac%2B-DuKZRmbCZ9jN50ZMNEo%3D&reserved=0 The funders did not play any role in the study design, data collection nor analysis, decision to publish, or preparation of the manuscript. JKP: 1R21HD103058-01 - National Institute of Child Health and Human Development; https://nam12.safelinks.protection.outlook.com/?url=https%3A%2F%2Fwww.nichd.nih.gov%2F&data=05%7C02%7C%7Ced-77110bc0e443962aea08de4438e926%7C58b3d54f16c942d3af081fcabd095666%7C1%7C0%7C639023208546348609%-7CUnknown%7CTWFpbGZsb3d8eyJFbX-B0eU1hcGkiOnRydWUsIlYiOiIwLjAuMDAw-MCIsIlAiOiJXaW4zMiIsIkFOIjoiTWFpbCIs-IldUIjoyfQ%3D%3D%7C0%7C%7C%7C%7C&s-data=%2FJip%2BKuBssTRd-vEJAMnYD%2BdWsNYr5XT-HO3gMcI%2BWe9A%3D&reserved=0 The funder did not play any role in the study design, data collection nor analysis, decision to publish, or preparation of the manuscript. JKP: No number - Doris Duke Charitable Foundation Caregivers at Carolina: Support for Physician Scientists; https://nam12.safelinks.protection.outlook.com/?url=https%3A%2F%2Fwww.doris-duke.org%2F&data=05%7C02%7C%7Ced-77110bc0e443962aea08de4438e926%7C58b3d54f16c942d3af081fcabd095666%7C1%7C0%7C639023208546496938%-7CUnknown%7CTWFpbGZsb3d8eyJFbX-B0eU1hcGkiOnRydWUsIlYiOiIwLjAuMDAwM-CIsIlAiOiJXaW4zMiIsIkFOIjoiTWFpbCIsIldUI-

feasible strategies for incorporating LIVEBORN Feedback into clinical care. We are now evaluating the effectiveness of LIVEBORN Feedback in a randomized control trial.

## Introduction

One million newborns die on their day of birth each year, accounting for one third of all newborn deaths [1]. Ninety percent of these deaths are from intrapartum-related events resulting in failure to breathe at birth (i.e., respiratory depression) [2]. Over 90% of these deaths occur in low and lower-middle income countries (LMICs) [3].

Basic resuscitation practices reduce death from respiratory depression [4]. Among these practices, bag-mask ventilation (BMV) has the greatest impact on mortality. To be effective, BMV must be timely. Resuscitation algorithms recommend initiating BMV of non-breathing newborns within 60 seconds after birth [5,6]. Delayed BMV increases the risk of death: for example, for every 30-second delay in BMV in a cohort of newborns in Tanzania, the risk of death or prolonged hospital admission increased by 16% [7,8]. Effective BMV is also continuous, with ventilation recommended until spontaneous breathing begins.

A common strategy to improve basic resuscitation practices, including BMV, is simulation training using a resuscitation algorithm such as Helping Babies Breathe (HBB) [9]. However, such training alone cannot ensure effective BMV; as a result, initial reductions in perinatal mortality that result from HBB training are often not sustained. Resuscitation providers' knowledge and skill may decline over time, particularly when they do not use what they learned in HBB training with sufficient regularity to maintain the new knowledge and skill [9]. Scalable, complementary strategies, therefore, are required to ensure timely and continuous BMV.

There is strong scientific premise for improving BMV with feedback strategies. Feedback on cardiopulmonary resuscitation for health professionals has been shown to improve performance across a range of settings [10–19]. Two feedback strategies for improving performance of BMV during simulation have been shown to be effective: *real-time guidance* (feedback during practice) [20–25] and *debriefing* (feedback after practice) [26,27]. In high-income countries these strategies are typically implemented at the bedside using expert clinicians and detailed data on resuscitation events [20–27]. Feedback in LMICs may have similar and significant impact; however, such strategies must be adapted to such settings, where expert clinicians and detailed data may be limited. Mobile health (mHealth) technology could enable the implementation and evaluation of such feedback strategies in LMICs [28]. To date, none have been formally evaluated.

We developed LIVEBORN Feedback, an mHealth application (app) that provides real-time guidance and supports debriefing for newborn resuscitations, based on real-time documentation of the baby's condition and the birth attendant's actions. Using a mixed-methods approach, we developed then evaluated the feasibility of a strategy to incorporate LIVEBORN Feedback into clinical practice in the Democratic Republic of the Congo (DRC).

joyfQ%3D%3D%7C0%7C%7C%7C&sdata=x-uRfAUYn7q9QzpGzzTWeU4e69FnWucrvsNLqO-x1gX3M%3D&reserved=0; https://www.med.unc.edu/facultyaffairs/faculty-development/caregivers-at-carolina/ The funders did not play any role in the study design, data collection nor analysis, decision to publish, or preparation of the manuscript.

**Competing interests:** I have read the journal's policy and the authors of this manuscript have the following competing interests: KRB: Supported by the Office of The Director, National Institutes of Health to the IDeA States Pediatric Clinical Trials Network under award number UG1 OD024943 to University of Kansas Medical Center. JKP: Received research funding from the National Institute of Child Health and Human Development, the Laerdal Foundation, the Doris Duke Charitable Foundation, the Thrasher Foundation and the Gates Foundation; she is also the recipient of a Laerdal Global Health monetary gift to support her research. CLB: Received funding from the National Institutes of Health as well as travel support from the American Academy of Pediatrics and Laerdal Global Health.

## Materials and methods

### Study design and participants

We conducted a pilot study of LIVEBORN Feedback using an explanatory sequential mixed methods approach [29]. The study was conducted in two urban health facilities in Kinshasa, DRC, each with an approximate birth census of 3,500 per year. Both facilities are primarily staffed by midwives who provide basic newborn resuscitation as standard care. One facility was allocated to LIVEBORN's real-time guidance feature while the other facility was allocated to LIVEBORN's debriefing feature.

The study had two phases: strategy design and refinement, and pilot testing of the designed strategies. All midwives employed at the two facilities were eligible to participate if they provided newborn care at the time of birth as part of their regular employment. We aimed to recruit the vast majority of midwives at both facilities in order to facilitate consistent use of LIVEBORN Feedback.

### Description of LIVEBORN feedback

LIVEBORN is an mHealth app for newborn resuscitation that allows an observer to document clinical care and the status of the newborn during a resuscitation. In real time, the observer records their observations on a touchscreen by selecting a time-stamped button for a particular action and then de-selecting the button when the action is completed (see observation screen at top left of Fig 1) [30]. LIVEBORN also records the baby's heart rate from a heart rate meter called NeoBeat (Fig 1). In this study, we evaluated a new feedback functionality of LIVEBORN. LIVEBORN provides two types of feedback: real-time guidance and debriefing. Both types of feedback are generated using the data registered by the observer and the recorded heart rate. During a resuscitation, LIVEBORN Feedback provides real-time guidance as audio prompts. After a resuscitation, LIVEBORN Feedback supports providers in data-driven debriefing through a series of screens with feedback and discussion points relevant for a specific resuscitation case. The goal of LIVEBORN Feedback is to help providers follow the HBB algorithm with a focus on timely and continuous ventilation. In previous simulations using a neonatal manikin with midwives in the DRC, we found LIVEBORN Feedback to be sufficiently usable and feasible to merit further evaluation in this clinical pilot [31].

### Training in use of LIVEBORN feedback

We provided each facility with two tablet-sized computers for using LIVEBORN Feedback and ensured they had adequate supplies for resuscitation including NeoBeat. Midwives at both facilities were accustomed to using NeoBeat in their clinical practice due to on-going use from a prior clinical trial.

We oriented all participating midwives to LIVEBORN, including a guided tour of the app and its features. Midwives practiced recording resuscitation events using LIVEBORN by observing and documenting care captured in a variety of video-recorded resuscitations. Midwives then practiced observing resuscitations at their health facility

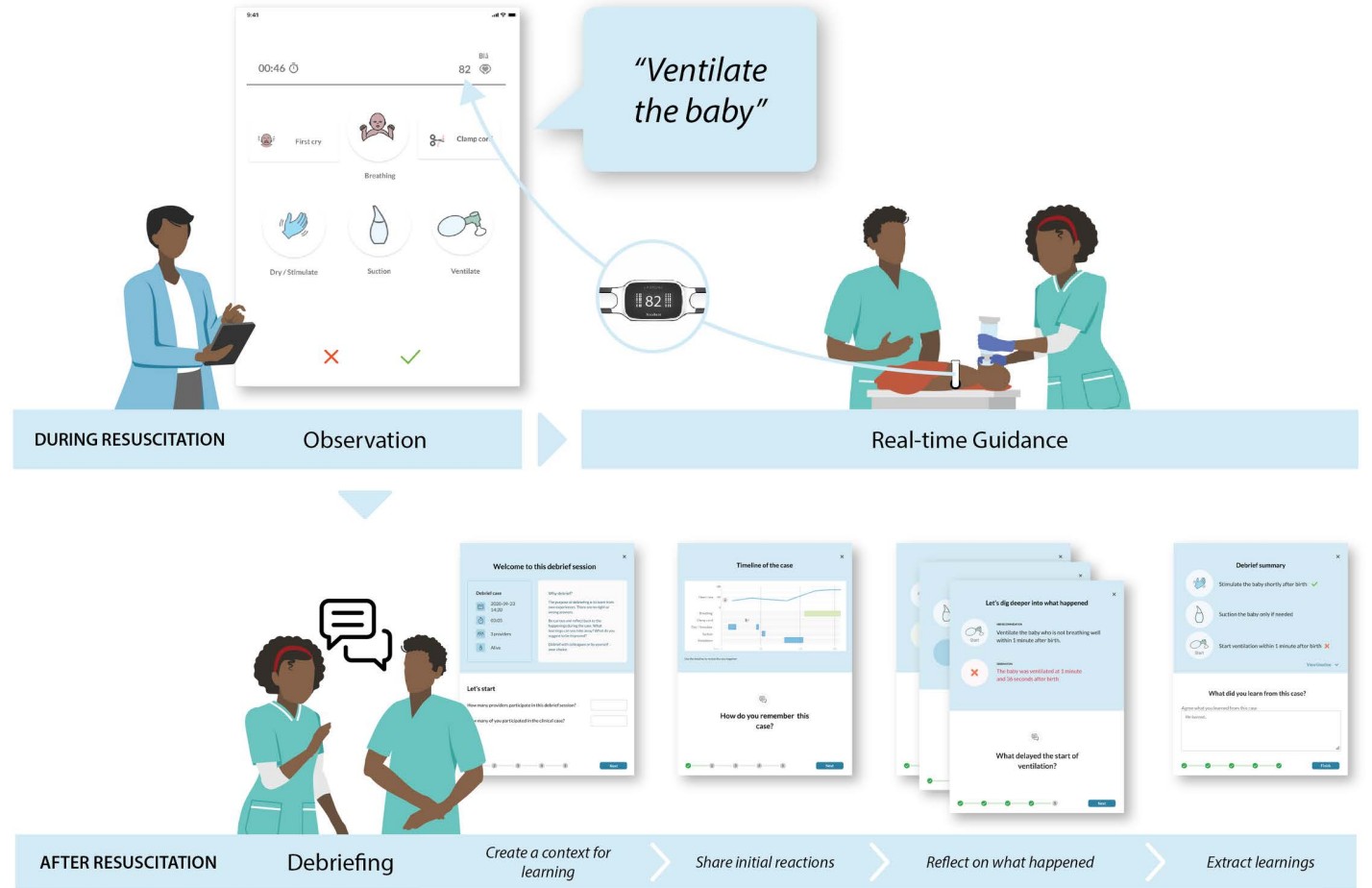

**Fig 1. LIVEBORN Feedback: mobile health application for newborn resuscitation.** LIVEBORN Feedback delivers real-time guidance during resuscitation and supports debriefing after resuscitation. An observer records the start and stop times of key provider actions during the resuscitation as well as when the baby first cries. A battery-operated heart rate meter called NeoBeat communicates the baby's heart rate to LIVEBORN via Bluetooth technology. LIVEBORN Feedback integrates the data from the observer and NeoBeat to deliver audio real-time guidance that prompts the provider to initiate the appropriate action to help the baby breathe (or, to stop harmful actions such as prolonged suctioning). Following a resuscitation, LIVEBORN Feedback supports the provider in debriefing by presenting objective data from the resuscitation, recommendations from an evidence-based resuscitation algorithm, and talking points to support reflection and discussion.

using LIVEBORN with real-time coaching from research staff. For example, research staff would provide tips for reducing errors in documentation with LIVEBORN both in the moment and following a case.

In the facility allocated to the real-time guidance feature, we conducted simulations with LIVEBORN Feedback's audio guidance to orient the midwives to the app interface and data entry procedures. In the facility allocated to the debriefing feature, we oriented midwives to LIVEBORN Feedback's debriefing screens. Because of limited experience with formal debriefing at this facility, we also conducted a half-day training with midwives at this facility. Developed by the study team, the debriefing training discussed the goals and steps for debriefing and emphasized the importance of reflexive practice and creating a culture of improving care. As part of this training, midwives practiced debriefing in several simulation exercises using LIVEBORN Feedback's debriefing feature. In anticipation of strategy design, we defined cases receiving BMV as the appropriate target for debriefing. However, midwives were free to debrief on additional cases that they deemed useful for learning.

## Strategy design and refinement

Following these trainings, we developed implementation strategies to incorporate LIVEBORN Feedback into clinical practice using a participatory research methodology called Trials of Improved Practices (TIPS) [32]. TIPS engages stakeholders in the design of behavior change activities through an iterative process involving strategy development, small-scale testing and rapid analysis. The goal for the strategy was use of LIVEBORN Feedback for at least 50% of facility births over the observation period. This design phase incorporated five elements described in Fig 2: 1) who observes births with LIVEBORN, 2) which births are observed, 3) a system for cleaning and charging NeoBeat, 4) a system for charging and storing the LIVEBORN tablet, and 5) a system for preparing all resuscitation equipment including NeoBeat and the tablet computer. For the facility allocated to debriefing, the strategy included a sixth element focused on a plan for debriefing—e.g., when debriefing takes place, who participates in debriefing, and which cases are debriefed. For each round of TIPS, we also planned how the strategy would be communicated to the midwives.

Each round of TIPS consisted of the following: (1) a strategy development session with the head nurse midwife and research team, (2) small-scale testing of the strategy in the health facility, and (3) rapid data analysis for further strategy refinement. While the target length of each round was one month, this could vary based on small-scale test results. During strategy development sessions, the head nurse midwife and research team discussed each element of the strategy and

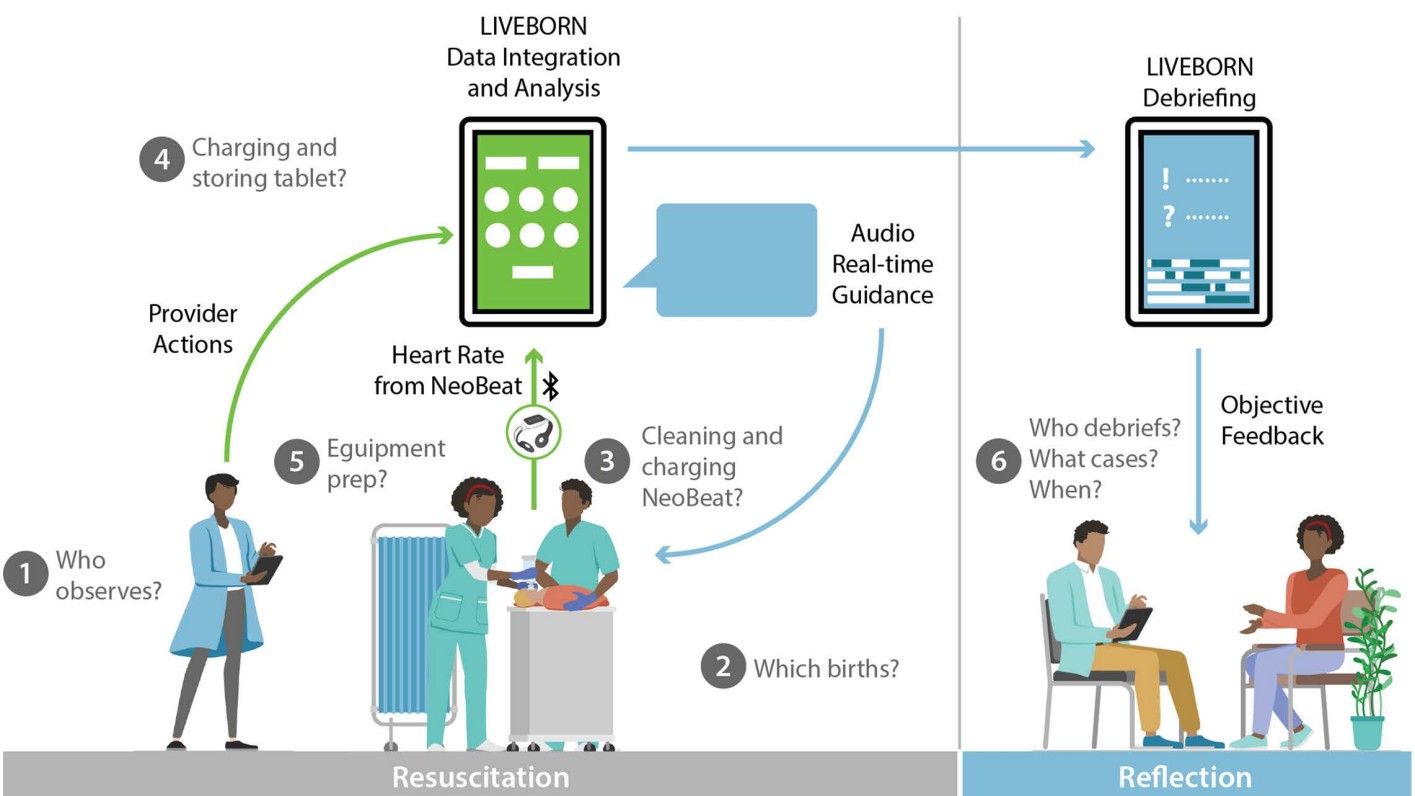

**Fig 2. Strategy design for implementation of LIVEBORN Feedback.** Using trials of improved practices, we developed an integrated strategy to incorporate LIVEBORN Feedback into clinical practice that included the following elements: 1) who observes births with LIVEBORN, 2) which births are observed, 3) a system for cleaning and charging NeoBeat, 4) a system for charging and storing the LIVEBORN tablet, and 5) a system for preparing all resuscitation equipment including NeoBeat and the tablet. At the facility allocated to LIVEBORN Feedback's debriefing feature, the strategy also included: 6) who debriefs, what cases are debriefed, and when debriefing occurs.

identified a feasible plan for implementation suitable to the local context. This strategy was then enacted by all midwife participants in the labor and delivery ward during small-scale testing. During this testing phase, research staff conducted weekly visits to directly observe the implementation strategy in clinical practice and to discuss with a convenience sampling of midwife participants. The research team rapidly analyzed these evaluation data to identify barriers and potential solutions to strategy implementation. Following rapid analysis, a new cycle began with strategy refinement by the head nurse midwife and research team. TIPS rounds continued at each facility until the facility reached the pre-determined feasibility target for LIVEBORN use (i.e., 50% of all deliveries).

## Pilot testing

At the conclusion of TIPS, a final strategy for integration of LIVEBORN Feedback into clinical practice was defined for implementation and evaluation in a three-month pilot test at each facility. During the pilot test, midwives at the facility began LIVEBORN implementation, based on the finalized strategy, within the context of routine clinical care. We collected data on the number of facility births each month, the number of LIVEBORN observations and the frequency of debriefing. At the end of the pilot, all midwives completed the System Usability Scale (SUS) to evaluate LIVEBORN's usability. At the facility allocated to real-time guidance, midwives additionally completed the Feasibility of Intervention Measure (FIM) [33,34].

The primary outcome for the pilot test was the feasibility of observing resuscitation care and entering relevant data into LIVEBORN, defined as the percentage of births observed using LIVEBORN. The *a priori* threshold for achieving feasibility was observing at least 50% of births. Secondary outcomes evaluated with *a priori* thresholds were as follows: 1) LIVEBORN usability, defined as a mean SUS score ≥68; 2) real-time guidance feasibility, defined as a median FIM score >12; and 3) debriefing feasibility, defined as debriefing for at least 50% of cases involving BMV. These thresholds were identified based on the prior literature and discussions with our research team and local collaborators. We used descriptive statistics (e.g., measures of centrality and dispersion for quantitative variables) to analyze SUS and FIM results (Microsoft Excel 2024, Version 16.86).

## Qualitative evaluation

We conducted two 90-minute focus group discussions (FGDs; one at each facility) with a convenience sample of six midwives per facility who had experience using LIVEBORN Feedback during TIPS and the pilot study. We excluded head nurse midwives from the FGDs to ensure open discussion. The quantitative results of the pilot study informed the FGD topics. In these FGDs, we explored 1) initial impressions of using LIVEBORN, 2) the process of observing births with LIVEBORN, 3) incorporation of LIVEBORN Feedback into clinical care, 4) any resulting changes in clinical care, 5) midwives' psychological safety as they used LIVEBORN Feedback, 6) recommendations for optimization of LIVEBORN Feedback, and 7) future use. A trained moderator (DI) conducted FGDs in French and Lingala, the local languages; both FGDs were audio-recorded with the consent of the participants and later transcribed. Lingala portions of the transcripts were translated into French, and data were coded in French.

We used qualitative content analysis to analyze data from the FGDs. ASM coded French transcripts using Lumivero, Nvivo (release 14.23.3). Two reviewers with proficiency in French (ASM and JKP) independently created English clusters of codes to identify patterns in the data, grouped clusters into themes, and determined final themes by consensus. The quantitative and qualitative data were merged following qualitative content analysis.

## Ethical approval

The study activities described above were approved by Institutional Review Boards at both the Kinshasa School of Public Health and the University of North Carolina at Chapel Hill (20–3414 and 22–0851). All participants provided informed written consent prior to enrolling in the study.

# Results

## Strategy design and refinement

For real-time guidance, strategy design involved 13 midwives and three rounds of TIPS (Table 1). The primary barrier encountered was insufficient staffing to observe 50% of births. The midwives and head nurse midwife determined to explore whether facility environmental health services staff could serve as potential observers due to their ready availability on the labor ward. Facility leadership approved of this plan, and a call system was developed to alert environmental health services staff for any impending birth. In the final plan, insufficient staffing for observing births was ultimately addressed by expanding the target population from high-risk births (selected in Round 1 of TIPS) to all births, and by adopting a three-pronged approach for observers that included tech-savvy midwives (those with experience using smartphones or touchscreen tablets), environmental health services staff and one full-time research staff member. The remaining elements of the final plan included: cleaning NeoBeat with other resuscitation equipment and, once clean, placing on the charger to be ready for its next use; maintaining the tablet in the acute care room, with the acute care nurse assigned to ensure it was charged; and the resuscitator preparing NeoBeat with the other resuscitation equipment and the observer preparing the LIVEBORN tablet before every birth.

For LIVEBORN debriefing, strategy design involved 18 midwives and six rounds of TIPS (Table 2). At this facility the primary barrier was also insufficient staff to observe 50% of births. The midwives tested several strategies for expanding the pool of observers including having trainees and a research staff member observe births; the midwives determined that collecting data with LIVEBORN during trainee-led resuscitations was a valuable teaching tool, and thus the final strategy for observers was limited to midwives and one research staff member. This facility also tested observing all births versus high-risk births, including the use of a checklist to identify high-risk births in round 3. As a focus on high-risk births did not result in sufficient observations, the final strategy focused on observing all births. Similar to the other facility, in the final strategy. NeoBeat was cleaned with other resuscitation equipment and placed on its charger to be ready for next use, and the resuscitator prepared NeoBeat with the other resuscitation equipment while the observer prepared the LIVEBORN tablet before every birth. Contrastingly, the tablet was stored in a wooden box with padlock that was kept near the NeoBeats; the wooden box was attached to a metal trolley so the observer could rest the tablet on the box during observations. The head nurse midwife ensured that the tablet was charged.

**Table 1. Strategy development during TIPS at the facility allocated to real-time guidance.**

| Element | Round 1 | Round 2 | Round 3 |
|---|---|---|---|
| **Birth observers** | Tech-savvy midwivess (n = 8) | Tech-savvy midwivess + EHS staff | Tech-savvy midwivess + EHS staff + study nurse |
| **Types of births observed** | High-risk births | All births | ▽ |
| **System for cleaning/ charging NeoBeat** | Clean with other resuscitation equipment per poster displayed; place clean devices on charger | ▽ | ▽ |
| **System for charging LIVEBORN tablet(s)** | Maintain in acute care room; acute care nurse ensures charged | ▽ | ▽ |
| **Resuscitation equipment preparation** | For every birth, resuscitator prepares NeoBeat with other equipment; when high-risk of resuscitation, observer prepares tablet | For every birth, resuscitator prepares NeoBeat with other equipment and observer prepares tablet | ▽ |
| **Cases observed with LIVEBORN, n(%)** | 29 (5)[1] | 95 (19) | 115 (65) |

▽ = no change.

EHS = environmental health services.

[1] 14% of observed cases included BMV compared to an anticipated rate of BMV for 3% of all births.

**Table 2. Strategy development during TIPS at the facility allocated to debriefing.**

| Element | Round 1 | Round 2 | Round 3 | Round 4 | Round 5 | Round 6 |
|---|---|---|---|---|---|---|
| **Birth observers** | All midwives | All midwives | All midwives | All mid-wives + study nurse | All mid-wives + medical trainees + study nurse[1] | All mid-wives + study nurse |
| **Types of births observed** | High-risk births[2] | All births | High-risk births with checklist[3] | All births | ▽ | ▽ |
| **System for cleaning/ charging NeoBeat** | Cleaning with other resuscitation equipment per poster displayed; place clean devices on charger | ▽ | ▽ | ▽ | ▽ | ▽ |
| **System for stor-ing/charging LIVEBORN tablet(s)** | Stored in wooden box with padlock kept near NeoBeats; charged by head nurse midwife | As per round 1, but with wooden box attached to metal trolley so observer can rest tablet on the box during observations | ▽ | ▽ | ▽ | ▽ |
| **Resuscitation equipment preparation** | For every birth, resuscitator prepares NeoBeat with other equipment; when high-risk of resuscitation, observer prepares tablet | For every birth, resuscitator prepares NeoBeat with other equipment and observer pre-pares tablet | ▽ | ▽ | ▽ | ▽ |
| **Plan for debriefing** | Debriefing of complex resuscitations[4] during morning meeting led by head nurse midwife with all midwives pres-ent for the shift participating | ▽ | ▽ | ▽ | ▽ | ▽ |
| **Cases observed with LIVE-BORN, n(%)** | 22 (15)[4] | 61 (29) | 59 (15)[5] | 185 (35) | 42 (70) | 64 (84) |

▽ = no change.

TIPS = trials of improved practices.

[1]After testing this strategy, midwives perceived it was better for them to observe and coach the trainee simultaneously. [2]High-risk criteria de-termined by the midwives, as follows: prolonged labor (>8 hours), meconium-stained amniotic fluid, premature rupture of membranes, polyhy-dramnios, oligohydramnios, preterm labor, maternal hypertension, maternal fever, abnormal fetal heart tones, or a laboring woman who appears fatigued in the assessment of the midwife. [3]High-risk criteria formally monitored by midwives using a paper checklist during labor; this checklist was used inconsistently and found to be burdensome so was eliminated for Round 4. [4]23% of observed cases included BMV compared to an anticipated rate of BMV for 3% of all births. [5]Results may have been influenced by unanticipated prolonged absence of the head nurse midwife during this round.

## Pilot testing

During pilot testing, LIVEBORN met the usability threshold overall with a median SUS score of 68 (Q1 65, Q3 78), and the percent of births observed with LIVEBORN met the feasibility threshold at both facilities (Table 3). At the real-time guidance facility, the median SUS score was 73 (Q1 68, Q3 83), and 74% of births were observed. LIVE-BORN real-time guidance also met the feasibility threshold with a median FIM score of 16 (Q1 16, Q3 16). At the debriefing facility, the median SUS score of 65 (Q1 60, Q3 72) fell below the *a priori* threshold for usability although the feasibility threshold was met with 67% of births being observed. Debriefings were conducted on three of the seven BMV cases during the pilot study, which was below the feasibility threshold of 50% of BMV cases debriefed. Of note, six cases without BMV were also debriefed with a time to first breath in those cases ranging from 49-163 seconds after birth. An average of three midwives (range: 1–4) attended each debriefing, and each midwife attended an average of one debriefing per month.

**Table 3. Implementation outcomes from feasibility testing of LIVEBORN[1].**

| Implementation Outcomes | Result | *A Priori* Threshold (Justification) |
|---|---|---|
| *Usability* | | |
| **LIVEBORN app** *Median SUS score (quartiles)* | | |
| **Overall, n = 29 midwives surveyed** | 68 (65, 78) | SUS score ≥68 (threshold for usability) |
| **Facility A, n = 13 midwives surveyed** | 73 (68, 83) | |
| **Facility B, n = 16 midwives surveyed** | 65 (60, 72) | |
| *Feasibility* | | |
| **Observing births with LIVEBORN** *Births observed (percent of all births)* | | |
| **Facility A, n = 968 births** | 712 (74) | ≥50% births observed (minimum to maintain power for RCT) |
| **Facility B, n = 292 births** | 195 (67) | |
| **Real-time guidance with LIVEBORN (Facility A only)** *Median FIM score (quartiles)* *N = 13 midwives surveyed* | 16 (16, 16) | FIM score >12 (threshold salient for adoption) |
| **Debriefing with LIVEBORN (Facility B only)** *BMV cases debriefed (percent of all BMV cases)* *N = 7 BMV cases* | 3 (42) | ≥50% of BMV cases debriefed (minimum uptake for impact) |

SUS = System Usability Scale; FIM = Feasibility of Intervention Measure; BMV = bag-mask ventilation; RCT = randomized controlled trial.

[1]Pilot study dates: Facility A 5/23/22–8/31/22; Facility B 7/18/22–8/31/22.

## Midwife perceptions about LIVEBORN

In FGDs conducted one year after the pilot study with midwives from both facilities (Table 4), the midwives were positive in their perceptions: they felt that LIVEBORN Feedback could save lives and they expressed interest in ongoing use of the app. Midwives described experiencing a learning curve observing cases with LIVEBORN. In particular, they noted technology challenges including touchpad issues and lack of familiarity with using the tablet. One midwife mentioned having to remember to wear her glasses to use the touchpad. Touchpad issues were resolved by practicing with the tablet and sharing tips on its use.

Midwives working with the real-time guidance feature found the audio guidance to be helpful, with an appropriate level of detail. They did not have negative perceptions of how it delivered guidance. They perceived inconsistencies with the audio guidance and felt the prompts moved too quickly. As a result, they acknowledged that some prompts were followed while others were not. In particular, midwives described scenarios where they ignored the guidance to stop suctioning and move on to ventilation because they were confident the baby would cry after more suctioning. They recommended using real-time guidance for all livebirths since it cannot always be predicted which neonate will require resuscitation.

Midwives working with the debriefing feature described feeling discouraged by their mistakes but sought to improve. Debriefings offered a forum for learning, which allowed them to correct mistakes and increased their self-confidence. The midwives stated that LIVEBORN Feedback's debriefing prompts were easy to understand, but they sometimes questioned the prompts. As with real-time guidance, midwives described scenarios where they disagreed with the guidance to ventilate, pointing out the baby eventually cried with suctioning. Additionally, midwives emphasized the importance of everyone giving and receiving feedback with love and acceptance, and the importance of building a culture of improvement. They suggested debriefing for all neonates requiring resuscitation.

**Table 4. Illustration of qualitative content analysis with theme, example cluster and exemplary quote.**

| | Theme | Example Cluster | Exemplary Quote |
|---|---|---|---|
| **Real-time Guidance and Debriefing** | Midwives believed that LIVEBORN could save lives. | The tool helped us resuscitate the most depressed babies | "This instrument has saved many children." |
| | | | "The more we used it, the better we were able to move quickly." |
| | Midwives experienced a learning curve observing cases with LIVEBORN and were unable to observe all cases. | Could not observe all births with LIVEBORN | "When we delivered the baby, you can be drying him off and he cries out straight away, but the person observing isn't quick enough or doesn't master handling the tablet well…Sometimes when you cut the cord, instead of the person pressing down, they forget…so it was a bit of a mess at first." |
| | Midwives were interested in ongoing use of LIVEBORN; they suggested debriefing for all neonates requiring resuscitation and real-time guidance for all livebirths. | Interested in continued use of real-time guidance | "As we've evolved, we've noticed that some children, for example, whose mothers have also given birth well, can come out well and give a good first cry, after which there's no more crying. So if there's no tablet, it's difficult to start at that moment; but if there's already a tablet next to them, you can already continue because they're registered." |
| | | | "We would like you to bring it back to us even if we insult it." |
| **Real-time Guidance** | Despite perceived inconsistencies with the guidance, the audio guidance helped us [midwives]. | Tool was not distracting and helped us | "It reminded us what to do without disturbing us." |
| | | | "It didn't interfere with or delay our resuscitation practice." |
| | | | "It helped us a lot with resuscitation." |
| | Audio prompts did not feel like a reprimand. | Prompts were supportive to us | "The tool didn't stress us out, but rather comforted us." |
| | Midwives followed some prompts and not others. | Discounted prompts related to suctioning | "Sometimes you suction the child and it tells you 'stop suctioning if there are no more secretions' while you feel that the child still has secretions, so you say to yourself if I keep suctioning the cry will come out. But over there the voice tells you 'Stop suctioning! Stop suctioning! Switch to ventilation' when you know that with suction alone, he'll cry out." |
| | Prompts were appropriately detailed but felt fast. | Prompts too fast | "For me it [prompts] appeared early/very early as the other colleagues said, you're still suctioning but it's telling you to stop suctioning to move on to the next stage."<br>"He [prompts] talked through all the steps and that helped us a lot, but we just had to add clamping the cord that wasn't there." |
| **Debriefing** | The head nurse midwife facilitated debriefing during morning meetings. | Debriefed with colleagues, even those not present for the case | "Every day, the head nurse midwife keeps track of the cases in the tablets, and if there are any problems, she calls you to discuss them." |
| | Midwives were both discouraged by their mistakes and encouraged to improve. | Sad but still wanting to participate | "It was a bit sad and embarrassing." |
| | | | "If there's a remark that weighs heavily, I wouldn't say it offended me, but it makes me sad. Why didn't I follow up on the comments made by colleagues? So being with colleagues in a meeting, having comments but doing something contrary to what was said in front of colleagues and managers, makes us a little sad, but it also encourages us to wake up and do better in the future." |
| | LIVEBORN taught midwives to correct their mistakes and increased self-confidence. | More orderly resuscitations with respect for the golden minute | "During the debriefings at the beginning, we had to make the comparison with our previous resuscitations. In our previous resuscitations there were steps that we didn't respect; there were archaic maneuvers that we used, for example you tap the child on the back, you turn the child…We found that some of the maneuvers we were practicing were still causing the child suffering…[Now] there are resuscitation materials; you place it [NeoBeat] and you follow its beats as you resuscitate it [the baby]. That's what made it possible for us to leave the old practices that we shouldn't be using anymore." |
| | | | "Our mistakes were identified, particularly the golden minute." |
| | LIVEBORN prompts were easy to understand but sometimes perceived as erroneous. | Inconsistencies with debriefing prompts at times | "Debriefing says to ventilate, but you find there was no problem with the resuscitation." |

## Discussion

The purpose of this study was to test the feasibility of using an mHealth app, LIVEBORN, to collect observational data during newborn resuscitations and to use these data to guide clinical practice using two approaches. Partnering with front-line workers in an iterative fashion, we successfully identified feasible strategies to incorporate LIVEBORN Feedback into clinical practice. Midwives readily adopted systems for maintaining the technology (tablets, NeoBeat) but were challenged by the need to observe resuscitations to use LIVEBORN Feedback. Overall, midwives found LIVEBORN to be usable. Real-time guidance met all pre-specified criteria for feasibility. While debriefing was also viewed favorably and used in some cases beyond the pre-established criteria for which births to debrief, it did not reach one of two feasibility thresholds designated before the study. Nevertheless, these results are highly promising and provide important formative data for future scale-up in similar settings.

The requirement of an observer to collect data during a resuscitation in order to use LIVEBORN Feedback is a challenge for scale-up in LMICs. Given understaffing of healthcare providers in the two study facilities, a common occurrence in low-resource settings, identifying observers to use LIVEBORN was the most challenging part of the strategy. In addition, limited prior exposure of participants to smartphones and/or tablets was also a barrier to observing births. To address these issues, we tested novel approaches for identifying intra-facility observers including nurse trainees and non-clinical staff. While these approaches augmented our ability to observe births with LIVEBORN, we ultimately needed to introduce an observer external to the facility (a research study nurse) to achieve our goal. It is also important to note that LIVE-BORN Feedback is dependent upon accurate data describing the condition of the newborn, particularly breathing, and actions of providers. Our training process for observers of LIVEBORN was intensive but essential. While real-time guidance requires real-time data entry, it is possible that debriefing could be accomplished with data documented shortly after the resuscitation. However, such posthoc entry may be subject to recall biases that could reduce the effectiveness of this feedback approach [35].

Alternative approaches to observation may also hold promise. For example, technology-based strategies, including automatic interpretation of video footage using artificial intelligence and automated detection of BMV via sound, could eliminate the need for an observer. Our team is currently working to develop this capacity, as it seeks to enhance the usability of LIVEBORN. At present, however, an auditing approach may be best where LIVEBORN is used to review care occasionally rather than to collect data continuously.

Midwives found LIVEBORN real-time guidance feasible to implement clinically. Although midwives initially were intimidated and overwhelmed by the technology, particularly using a tablet, they came to accept and integrate the guidance into their care of newborns. While midwives had a positive impression of the guidance overall and were interested in future use of LIVEBORN Feedback, disagreements with some of the guidance pose concern for whether real-time guidance will change behavior. For example, firmly held belief that suctioning rather than BMV leads to crying could pose barriers to improving care with real-time guidance. There was also a sensation of feeling rushed by the real-time guidance despite the timing of the advice being appropriate for HBB adherence. As has been noted in a prior study on video telemedicine for newborn resuscitation, the adoption of guidance is linked to the perceived benefit [36]. Beyond feasibility, instilling confidence in the guidance may be key to changing behavior. Ongoing professional education may also alter response to guidance.

The only feasibility criterion that was not met during the pilot test was debriefing for at least 50% of BMV cases. We counted a case as having been debriefed once the midwives clicked the "finish" button at the end of the LIVEBORN debriefing screens. We suspect our quantitative results underestimate the actual number of cases debriefed; anecdotally, midwives debriefed more frequently but may not have consistently closed out of the final screen. The time involved to debrief may be a barrier and may have contributed to lower SUS scores in the debriefing facility compared to the real-time guidance facility.

Although we were below the debriefing target, midwives selected many additional cases they felt were important to debrief which did not involve BMV, an indication of the perceived value of the practice. (Despite prior data in these health facilities indicating that providers fail to initiate ventilation in 80% of the cases in which it is indicated, [37] cases for debriefing may be better selected based on time to breathing rather than receipt of BMV. Each midwife attended an average of one debriefing per month. Although data about the frequency of debriefing is limited, low exposure could influence time to practice change or practice change at all. Additionally, our qualitative data also illustrates the tension between discouragement when discussing mistakes and motivation to improve. Building a culture of psychological safety may be important for debriefing to be effective, and culture change takes time. Shared responsibility for resuscitations has been suggested as one way to avoid personal blame and learn from mistakes, as noted by a study with midwives in Nepal [38].

Our iterative approach to strategy design was an innovation of the study. Borrowing from the field of nutrition, [32] we used participatory research methodology called TIPS to engage stakeholders in the design of activities that involved behavior change. The iterative nature of this process permitted midwives to trial and rapidly improve strategies. This methodology allowed us to identify barriers to implementation of LIVEBORN Feedback and to develop strategies to overcome the barriers that were suitable for the context. In particular, we noted that unfamiliarity with technology can be a barrier to its implementation; attention to systems that make technology adoption easier (such as storage, cleaning, and charging) were an important part of TIPS. Implementation scientists and program implementers, particularly in the field of mHealth, may find this methodology suitable methodology for introducing other innovations in otherwise technology-naïve, clinical environments.

Strengths of our studies include an emphasis on stakeholder engagement and our use of a mixed-methods approach to evaluate feasibility. We also note important limitations. Our study was limited by the small sample size (in terms of the number of facilities and the number of participating midwives) and the short duration of our pilot study. Our qualitative analysis is limited by recall bias since data were collected well after the pilot phase. Finally, our findings reflect the feasibility of incorporating LIVEBORN Feedback into urban health facilities in Kinshasa and may not be generalizable to other health facility settings in low-resource environments. Despite these limitations, our results provide an important foundation for future work, providing critical formative evidence to guide further scale-up and research.

## Conclusion

We identified context-specific, feasible strategies for incorporating LIVEBORN Feedback into clinical care in two facilities in the DRC. There is an ongoing randomized control trial to investigate the important question of LIVEBORN Feedback's effectiveness [39]. Future studies on implementation of LIVEBORN Feedback should consider how to build trust in the guidance provided and how to foster a facility culture of improvement.

## Supporting information

**S1 File. The LIVEBORN study: An integrated mhealth strategy to improve newborn resuscitation in low- and lower middle-income countries protocol version 3.0.**
(PDF)

## Author contributions

**Conceptualization:** Carl L. Bose, Helge Myklebust, Benjamin H. Chi, Eric M. Mafuta, Jackie K. Patterson.

**Formal analysis:** Amy S. Mackay, Jackie K. Patterson.

**Funding acquisition:** Jackie K. Patterson.

**Investigation:** Daniel K. Ishoso, Eric M. Mafuta.

**Methodology:** Amy S. Mackay, Carl L. Bose, Patricia P. Gomez, Jackie K. Patterson.

**Project administration:** Daniel K. Ishoso, Eric M. Mafuta, Jackie K. Patterson.

**Resources:** Kourtney R. Bettinger, Ingunn A. Haug.

**Supervision:** Jackie K. Patterson.

**Visualization:** Ingunn A. Haug.

**Writing – original draft:** Kourtney R. Bettinger, Amy S. Mackay, Jackie K. Patterson.

**Writing – review & editing:** Daniel K. Ishoso, Amy S. Mackay, Carl L. Bose, Patricia P. Gomez, Ingunn A. Haug, Helge Myklebust, Benjamin H. Chi, Eric M. Mafuta, Jackie K. Patterson.

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
