## [Decision Letter · Decision Letter 0]

4 Apr 2025

Dear Dr. Patterson,

Thank you for submitting your manuscript to PLOS ONE. After careful consideration, we feel that it has merit but does not fully meet PLOS ONE’s publication criteria as it currently stands. Therefore, we invite you to submit a revised version of the manuscript that addresses the points raised during the review process.

We look forward to receiving your revised manuscript.

Kind regards,

Ju Ok Park

Academic Editor

PLOS ONE

Journal Requirements:

Additional Editor Comments:

Please consider the reviewer's comments and revise your manuscript accordingly.

Reviewers' comments:

Reviewer's Responses to Questions

**Comments to the Author**

1. Is the manuscript technically sound, and do the data support the conclusions?

Reviewer #1: Yes

Reviewer #2: Yes

2. Has the statistical analysis been performed appropriately and rigorously?

Reviewer #1: Yes

Reviewer #2: Yes

3. Have the authors made all data underlying the findings in their manuscript fully available?

Reviewer #1: Yes

Reviewer #2: Yes

4. Is the manuscript presented in an intelligible fashion and written in standard English?

Reviewer #1: Yes

Reviewer #2: Yes

Reviewer #1: This study introduces the mobile health application LIVEBORN as an approach to inducing effective bag-masking, based on the importance of bag-mask ventilation and the decline in training skills for it,

It also presents an implementation strategy for incorporating liveborn feedback into clinical practice using a participatory research methodology called Trials of Improved Practices (TIPs) to reach a target system usability scale and feasibility. Despite limitations, such as midwives sometimes not agreeing to ventilatory guidance and bag-mask ventilation case reflections not reaching feasibility figures, the study identifies feasible strategies for incorporating liveborn feedback into clinical care.

Mobile health applications are an effective means of improving the quality of resuscitation procedures in areas where medical and human resources are limited. However, its implementation is not easy in areas where it is actually needed. The paper also describes a process to measure feasibility and usability as a pilot study and to improve to a target point. The target setting is also appropriate. It is an important contribution to improving the quality of neonatal resuscitation in areas with limited medical and human resources.

Although the debriefing on the bag-mask ventilation cases has not reached a feasibility figure, rather, the six cases without BMV were also debriefing. As the authors discuss, the item setting may be best selected based on time to breathing.

The authors' meticulous research and deep insights will greatly advance our understanding of this field.

Reviewer #2: Thank you for the opportunity to review the manuscript titled, Mobile Health Supported Real-Time Guidance and Debriefing for Newborn Resuscitation: A Pilot Study of LIVEBORN Feedback. This paper presents new evidence for application of a real-time guidance tool for neonatal resuscitation in LMICs. The need for improving neonatal outcomes in LMICs and supporting midwives during reception of the newborn reflects the importance of LIVEBORN.

There are a few suggestions for the authors to consider before publication is considered, as follows:

Suggest minor editing to improve sentence structure and punctuation.

Introduction

p. 4. The introduction states “… HBB training are often not sustained due to a decline in knowledge and skill over time.” This sentence is not clear in relation to who HBB training is relevant to. For example, are you referring to midwives? Why is knowledge and skill related to BMV declining over time? Are T-piece devices available in these settings?

p. 5. What type of mixed methods approach was used? Please define and provide a reference to enhance this section.

Materials and Methods

Study Design and Participants

p. 5. Sampling strategy should be noted here. Was purposive sampling of participants used based on midwives who have specialised knowledge of resuscitation?

p. 5. What was the target recruitment number of midwives?

p. 5. A section on the study sites could have been added here. For example, how many births were recorded at each site per year? What level of capability/facilities available to support neonatal care at each site? Level of staffing at each facility.

Description of LIVEBORN Feedback

p. 6. More information needed on LIVEBORN usability would be beneficial. Does the observer manually type (free text) in observations during a real-time resuscitation or are there pre-populated fields and drop-down boxes to minimise data entry errors?

p. 6-7. Is the ongoing practice of prolonged suctioning linked to a decline in knowledge and skills over time (from p. 4)? It is not clear why this practice remains despite simulation training.

Training in Use of LIVEBORN Feedback

p. 7. You emphasized the importance of creating a culture of improving care. This sentence fails to acknowledge the importance of health professionals being reflexive practitioners. Suggest some mention of reflexive practice, which in turn influences culture and the provision of care.

Strategy Design and Refinement

More information on the methodology, Trials of Improved Practices, would enhance this section. Currently, this methodology is not well explained or justified when using a MM approach.

p. 8. “using a participatory research methodology called Trials of Improved Practices (TIPS)”. This sentence requires editing to improve readability. The first word, using, is not capitalised.

If over 50% of facility births were targeted, how many births does this equate to?

During the testing phase, what barriers were identified to implementation of the strategy? What changes were made in response to identified barriers?

Qualitative Evaluation

p. 10. Convenience sampling was used. Would purposive sampling be relevant considering midwives possess specialist knowledge on resuscitation?

pp. 10-11. Were all midwives who participated in the focus groups equal in terms of position status? It is not clear here if any midwife participants had higher responsibilities than the others with the potential for power imbalances existing during the focus groups.

Results

Strategy Design and Refinement

p. 12. Environmental health services staff have not been mentioned prior to now. Who are these staff? A description and recruitment strategy would be beneficial.

p. 12. “ … the nurse assigned to acute care ensured that it was charged”. Apologies if I missed this, but aren’t the participants midwives? If so, why are nurses included in the results? Is this nurse a research study nurse? Nurses in the acute care should be included in the methods if involved in the study. Were they recruited and PICF provided?

p. 12. Table 1 Who are skilled birth attendants? Are these midwives? There is some inconsistency with titles if they are the same. If not, please clarify each role.

p. 12. Table 1 Who observes births and Which births are observed are questions. Insert question marks or reword, for example, Birth observer and Observed birth type. Similarly, please edit questions in Table 2.

p. 12. Strategy design involving recruitment target for midwives or observed resuscitation activities should be stated in methods (recruitment).

p. 13. Legend requires punctuation between numbers 1-5.

p. 13. Checklist was found to be burdensome. In view of these burdensome features, what modifications have been made for the RCT?

p. 18. Nurse trainees and non-clinical staff as intra-facility observers were introduced in the discussion. These staff should have been introduced in the methods section, including recruitment strategies.

p. 18. Data documented shortly after the resuscitation highlights a limitation with LIVEBORN. What adaptations have been made to ensure post-resuscitation documentation is minimised or avoided for the proposed RCT?

p. 19. You state, “beyond feasibility, instilling confidence in the guidance may be key to changing behavior”. Might ongoing professional education of midwives be key to improving resuscitation techniques and hence, their behaviour?

p. 20. Edit out additional comma, “Given prior data in these health facilities indicating that providers fail to initiate ventilation in 80% of the cases in which it is indicated,, …”

p. 20. Additional cases were added to the debrief and results: Wasn’t there an eligibility for BMV debrief to ensure additional and irrelevant cases did not conflate your results?

p. 21. The use of mixed methods for analysis was not adequately linked to the theoretical approach of TIPS. At what points were the quantitative and qualitative data merged? Did the results of the quantitative data inform the topics for focus group discussion?

p.21. Limitations are clearly presented and provide direction for future implementation of LIVEBORN.

**Do you want your identity to be public for this peer review?** For information about this choice, including consent withdrawal, please see our Privacy Policy

Reviewer #1: No

Reviewer #2: **Yes: ** Melissa Blake

---

## [Author Response · Author response to Decision Letter 1]

20 May 2025

Thank you for the opportunity to review the manuscript titled, Mobile Health Supported Real-Time Guidance and Debriefing for Newborn Resuscitation: A Pilot Study of LIVEBORN Feedback. This paper presents new evidence for application of a real-time guidance tool for neonatal resuscitation in LMICs. The need for improving neonatal outcomes in LMICs and supporting midwives during reception of the newborn reflects the importance of LIVEBORN. There are a few suggestions for the authors to consider before publication is considered, as follows:

Suggest minor editing to improve sentence structure and punctuation.

Introduction

p. 4. The introduction states “… HBB training are often not sustained due to a decline in knowledge and skill over time.” This sentence is not clear in relation to who HBB training is relevant to. For example, are you referring to midwives? Why is knowledge and skill related to BMV declining over time? Are T-piece devices available in these settings?

Response: HBB training is relevant to any provider who performs newborn resuscitation in a low-resource environment (i.e., where chest compressions, epinephrine and oxygen are not standardly used in the delivery room). The HBB literature has demonstrated that knowledge and skill related to BMV decline over time; the infrequent use of BMV in clinical care, compounded in facilities with low delivery volumes, may contribute to this decline. T-piece devices are not typically available in settings where HBB is used; in the settings in which this evaluation took place, T-piece devices were not available. We clarified this sentence in the introduction as follows:

“Resuscitation providers’ knowledge and skill may decline over time, particularly when they do not use what they learned in HBB training with sufficient regularity to maintain the new knowledge and skill.” (Introduction, paragraph 3, page 4)

p. 5. What type of mixed methods approach was used? Please define and provide a reference to enhance this section.

Response: We used an explanatory sequential mixed methods approach. We added this description with a reference to the first sentence of the materials and methods section.

Materials and Methods: Study Design and Participants

p. 5. Sampling strategy should be noted here. Was purposive sampling of participants used based on midwives who have specialised knowledge of resuscitation?

Response: All midwives who provide newborn care at the facilities were eligible to participate. During TIPS, we used convenience sampling to discuss the strategy with midwives. We have added this detail about convenience sampling to the methods (Materials and Methods: Strategy Design and Refinement, paragraph

2, page 10).

p. 5. What was the target recruitment number of midwives?

Response: We aimed to recruit the vast majority of midwives at both facilities in order to facilitate consistent use of LIVEBORN Feedback. We have added this sentence to the materials and methods section, study design and participants, second paragraph on page 6.

p. 5. A section on the study sites could have been added here. For example, how many births were recorded at each site per year? What level of capability/facilities available to support neonatal care at each site? Level of staffing at each facility.

Response: We addressed these questions with the following addition:

“The study was conducted in two urban health facilities in Kinshasa, DRC, each with an approximate birth census of 3,500 per year. Both facilities are primarily staffed by midwives who provide basic newborn resuscitation as standard care.” (Materials and Methods: Study Design and Participants, paragraph 1, page 5)

Materials and Methods: Description of LIVEBORN Feedback

p. 6. More information needed on LIVEBORN usability would be beneficial. Does the observer manually type (free text) in observations during a real-time resuscitation or are there pre-populated fields and drop-down boxes to minimise data entry errors?

Response: We added the following description:

“In real time, the observer records their observations on a touchscreen by selecting a time-stamped button for a particular action and then de-selecting the button when the action is completed (see observation screen at top left of Figure 1).” (Materials and Methods: Description of LIVEBORN Feedback, paragraph 1, page 6)

p. 6-7. Is the ongoing practice of prolonged suctioning linked to a decline in knowledge and skills over time (from p. 4)? It is not clear why this practice remains despite simulation training.

Response: No. We think the ongoing practice of prolonged suctioning is due to lack of adoption of limited suctioning following HBB training. We have previously demonstrated this in a study of practices following HBB training in the DRC (Patterson et al., Children 2023).

Materials and Methods: Training in Use of LIVEBORN Feedback

p. 7. You emphasized the importance of creating a culture of improving care. This sentence fails to acknowledge the importance of health professionals being reflexive practitioners. Suggest some mention of reflexive practice, which in turn influences culture and the provision of care.

Response: Reflexive practice is a good term to capture what the debriefing training encouraged. This was added to the manuscript (Materials and Methods: Training in Use of LIVEBORN Feedback, paragraph 3, page 8).

Materials and Methods: Strategy Design and Refinement

More information on the methodology, Trials of Improved Practices, would enhance this section. Currently, this methodology is not well explained or justified when using a MM approach.

Response: We added the following description of TIPS to the methodology:

“TIPS engages stakeholders in the design of behavior change activities through an iterative process involving strategy development, small-scale testing and rapid analysis.” (Materials and Methods: Strategy Design and Refinement, paragraph 1, page 8)

p. 8. “using a participatory research methodology called Trials of Improved Practices (TIPS)”. This sentence requires editing to improve readability. The first word, using, is not capitalised.

Response: Thank you for catching this typo, which has been corrected.

If over 50% of facility births were targeted, how many births does this equate to?

Response: The census varied between the two facilities, and fluctuated month to month; the length of each TIPS cycle also varied. Thus, our target was a percentage rather than a fixed number of births. We calculated this target dynamically based on the denominator of births at the facility during a given round of TIPS. The final row of both Table 1 and Table 2 (i.e., “Cases observed with LIVEBORN”) provides the number of births observed for each TIPS cycle and the percentage of total births this number represented.

During the testing phase, what barriers were identified to implementation of the strategy? What changes were made in response to identified barriers?

Response: The primary barrier encountered at both facilities was insufficient staffing to observe births. We have added more detail about changes made in response to this barrier in paragraphs 1 & 2 of the results section on pages 12-14.

Materials and Methods: Qualitative Evaluation

p. 10. Convenience sampling was used. Would purposive sampling be relevant considering midwives possess specialist knowledge on resuscitation?

Response: All study participants were midwives and thus had specialized knowledge on newborn resuscitation. We chose convenience sampling because all participants had similar exposure to our intervention.

pp. 10-11. Were all midwives who participated in the focus groups equal in terms of position status? It is not clear here if any midwife participants had higher responsibilities than the others with the potential for power imbalances existing during the focus groups.

Response: We excluded head nurse midwives from the FGDs to ensure open discussion. We added this detail to the manuscript (Materials and Methods: Qualitative Evaluation, paragraph 1, page 11).

Results: Strategy Design and Refinement

p. 12. Environmental health services staff have not been mentioned prior to now. Who are these staff? A description and recruitment strategy would be beneficial.

Response: We added more detail about the incorporation of environmental health services staff in the first paragraph of the results section on page 12.

p. 12. “ … the nurse assigned to acute care ensured that it was charged”. Apologies if I missed this, but aren’t the participants midwives? If so, why are nurses included in the results? Is this nurse a research study nurse? Nurses in the acute care should be included in the methods if involved in the study. Were they recruited and PICF provided?

Response: This was a facility-level implementation study. The University of North Carolina Institutional Review Board and the Kinshasa School of Public Health Institutional Review Board considered midwives study participants as research staff directly interacted with them to develop the implementation strategy, and collected data about their care. As such, all midwives participating in the study gave informed consent. Other facility staff involved in the implementation were not considered research subjects; they were engaged by the midwives themselves to assist in the implementation, not by research staff.

p. 12. Table 1 Who are skilled birth attendants? Are these midwives? There is some inconsistency with titles if they are the same. If not, please clarify each role.

Response: The skilled birth attendants are midwives. The table has been changed to eliminate inconsistency.

p. 12. Table 1 Who observes births and Which births are observed are questions. Insert question marks or reword, for example, Birth observer and Observed birth type. Similarly, please edit questions in Table 2.

Response: Questions were reworded as suggested.

p. 12. Strategy design involving recruitment target for midwives or observed resuscitation activities should be stated in methods (recruitment).

Response: We added the following to the methods section:

“We aimed to recruit the vast majority of midwives at both facilities in order to facilitate consistent use of LIVEBORN Feedback.” (Materials and Methods: Study Design and Participants, paragraph 2, page 6)

p. 13. Legend requires punctuation between numbers 1-5.

Response: Punctuation has been added.

p. 13. Checklist was found to be burdensome. In view of these burdensome features, what modifications have been made for the RCT?

Response: The checklist was eliminated, as the final successful strategies in this pilot involved attempting to observe all births rather than restricting to high-risk cases.

Discussion

p. 18. Nurse trainees and non-clinical staff as intra-facility observers were introduced in the discussion. These staff should have been introduced in the methods section, including recruitment strategies.

Response: We added more detail about nurse trainees and non-clinical staff in the results section, paragraphs 1 & 2 on pages 12-13. As these individuals were part of the strategy, but not considered research participants, we did not include them in the methods. See additional explanation in response to #16 above.

p. 18. Data documented shortly after the resuscitation highlights a limitation with LIVEBORN. What adaptations have been made to ensure post-resuscitation documentation is minimised or avoided for the proposed RCT?

Response: LIVEBORN was designed for real time data collection by an observer. Observers in this study only documented care in real-time. We have clarified this in the Materials and Methods (Description of LIVEBORN Feedback, paragraph 1, page 6) and Discussion (paragraph 2, page 19).

p. 19. You state, “beyond feasibility, instilling confidence in the guidance may be key to changing behavior”. Might ongoing professional education of midwives be key to improving resuscitation techniques and hence, their behaviour?

Response: We have added this to the discussion:

“Ongoing professional education may also alter response to guidance.” (Discussion, paragraph 4, page 20)

p. 20. Edit out additional comma, “Given prior data in these health facilities indicating that providers fail to initiate ventilation in 80% of the cases in which it is indicated,, …”

Response: Completed. Thank you for identifying this typo.

p. 20. Additional cases were added to the debrief and results: Wasn’t there an eligibility for BMV debrief to ensure additional and irrelevant cases did not conflate your results?

Response: While the target for debriefing was cases with BMV, midwives were free to debrief on additional cases that they deemed useful for learning. This detail was added to the materials and methods (Training in Use of LIVEBORN Feedback, paragraph 3, page 8).

p. 21. The use of mixed methods for analysis was not adequately linked to the theoretical approach of TIPS. At what points were the quantitative and qualitative data merged? Did the results of the quantitative data inform the topics for focus group discussion?

Response: Quantitative and qualitative data were merged at the end of the study (after the qualitative data were analyzed). The results from the quantitative data did inform the topics for focus group discussion. These details were added to the Materials and Methods (Qualitative Evaluation, paragraphs 1 and 2, pages 11-12).

---

## [Decision Letter · Decision Letter 1]

23 Dec 2025

Mobile Health Supported Real-Time Guidance and Debriefing for Newborn Resuscitation:

A Pilot Study of LIVEBORN Feedback

PONE-D-24-35966R1

Dear Dr. Patterson,

We’re pleased to inform you that your manuscript has been judged scientifically suitable for publication and will be formally accepted for publication once it meets all outstanding technical requirements.

Kind regards,

Ju Ok Park

Academic Editor

PLOS ONE

Additional Editor Comments (optional):

Reviewers' comments:

Reviewer's Responses to Questions

**Comments to the Author**

Reviewer #2: All comments have been addressed

2. Is the manuscript technically sound, and do the data support the conclusions?

Reviewer #2: Yes

3. Has the statistical analysis been performed appropriately and rigorously?

Reviewer #2: Yes

4. Have the authors made all data underlying the findings in their manuscript fully available?

Reviewer #2: Yes

5. Is the manuscript presented in an intelligible fashion and written in standard English?

Reviewer #2: Yes

Reviewer #2: Thank you for your revisions. These revisions have improved the reader's understanding of the design using MM. I am happy to endorse the publication of this research. Well done on contributing new knowledge for health professionals when conducting neonatal resuscitation.

**Do you want your identity to be public for this peer review?** For information about this choice, including consent withdrawal, please see our Privacy Policy

Reviewer #2: **Yes: ** Melissa Blake

---

## [Editor Report · Acceptance letter]

PONE-D-24-35966R1

PLOS One

Dear Dr. Patterson,

I'm pleased to inform you that your manuscript has been deemed suitable for publication in PLOS One. Congratulations! Your manuscript is now being handed over to our production team.

Kind regards,

on behalf of

Mr Kindu Yinges Wondie

Academic Editor

PLOS One